# TiO_2_-Modified Montmorillonite-Supported Porous Carbon-Immobilized Pd Species Nanocomposite as an Efficient Catalyst for Sonogashira Reactions

**DOI:** 10.3390/molecules28052399

**Published:** 2023-03-06

**Authors:** Yuli Chen, Kailang Sun, Taojun Zhang, Jie Zhou, Yonghong Liu, Minfeng Zeng, Xiaorong Ren, Ruokun Feng, Zhen Yang, Peng Zhang, Baoyi Wang, Xingzhong Cao

**Affiliations:** 1Research Center of Advanced Catalytic Materials and Functional Molecular Synthesis, Zhejiang Key Laboratory of Alternative Technologies for Fine Chemicals Process, College of Chemistry and Chemical Engineering, Shaoxing University, Shaoxing 312000, China; 2Institute of High Energy Physics, Chinese Academy of Sciences, Beijing 100049, China

**Keywords:** Sonogashira reactions, heterogeneous catalysis, structure–performance relations, positron annihilation

## Abstract

In this study, a combination of the porous carbon (PCN), montmorillonite (MMT), and TiO_2_ was synthesized into a composite immobilized Pd metal catalyst (TiO_2_-MMT/PCN@Pd) with effective synergism improvements in catalytic performance. The successful TiO_2_-pillaring modification for MMT, derivation of carbon from the biopolymer of chitosan, and immobilization of Pd species for the prepared TiO_2_-MMT/PCN@Pd^0^ nanocomposites were confirmed using a combined characterization with X-ray diffraction (XRD), Fourier transform infrared spectroscopy (FTIR), N_2_ adsorption–desorption isotherms, high-resolution transition electron microscopy (HRTEM), X-ray photoelectron spectroscopy (XPS), and Raman spectroscopy. It was shown that the combination of PCN, MMT, and TiO_2_ as a composite support for the stabilization of the Pd catalysts could synergistically improve the adsorption and catalytic properties. The resultant TiO_2_-MMT_80_/PCN_20_@Pd^0^ showed a high surface area of 108.9 m^2^/g. Furthermore, it exhibited moderate to excellent activity (59–99% yield) and high stability (recyclable 19 times) in the liquid–solid catalytic reactions, such as the Sonogashira reactions of aryl halides (I, Br) with terminal alkynes in organic solutions. The positron annihilation lifetime spectroscopy (PALS) characterization sensitively detected the development of sub-nanoscale microdefects in the catalyst after long-term recycling service. This study provided direct evidence for the formation of some larger-sized microdefects during sequential recycling, which would act as leaching channels for loaded molecules, including active Pd species.

## 1. Introduction

Pd-based catalysts play important roles in many chemical transformations. In most applications, the homogeneous Pd catalyst contains not only the active Pd species but also various necessary ligands. Such homogeneous catalysis processes often suffer the difficulties of separation of the catalysts and ligands, recovery of the catalysts, and purity of the products, which make the processes un-green and cost-effective [1,2,3,4]. Therefore, the immobilization of active Pd species on appropriate supports to develop heterogeneous catalysts has received increasing attention both in the academic and industrial fields [5,6].

Recently, a variety of supports, such as active carbon, zeolites, MOFs, magnetic materials, clay, inorganic oxides (such as Al_2_O_3_, SiO_2_, TiO_2_, etc.), and organic polymers, have been attempted to immobilize Pd species to create heterogeneous catalyst systems [7,8,9,10,11,12,13]. Among these supports, active carbon is one of the most extensively investigated Appendix A owing to its high adsorption, good chemical inertness, and satisfied chelation. However, most pores contained in the conventional activated carbons are at the microporous level (<2 nm), which limits the transfer efficiency of large-sized reactant and product molecules into and out of the pores [14,15]. Therefore, the introduction of additional larger-sized porous structures is required for the porous carbon supports. The most frequently used modification method is adding hard or soft templates to induce abundant meso-/macro-porous structure. For examples, after removing the templates of polyether and/or colloidal SiO_2_ using pyrolysis or HF etching [16,17,18,19], the prepared porous carbon-supported Pd catalysts show a high mesoporous structure with high catalysis efficiency and stability as applied in coupling reactions. However, the added templates are required to be removed completely, leading to a complicated and un-economical preparation process. Hybridizing with other mesoporous materials such as montmorillonite (MMT) has been proven as another effective strategy to improve the adsorption capacity [20,21,22,23].

MMT clay [24] is a kind of abundant clay mineral in nature with a layered porous structure. The MMT layer consists of a central alumina octahedron sheet and two silicon tetrahedral sheets. Al^3+^ and Si^4+^ within the lattice structure are easily substituted by other ions with lower valence such as Mg^2+^, resulting in negative charges on the MMT layers. Moreover, they are often compensated by exchangeable positive cations, such as Na^+^ and Ca^2+^. Driven by ion exchange or hydrogen–bonding interactions, many natural macromolecules can be easily intercalated into the MMT layers. After further hydrothermal carbonization, the natural macromolecules-derived porous carbon (PCN) can be well supported on the MMT matrices. For example, Pei et al. [25] prepared novel MMT-supported porous carbon (PCN) nanosphere adsorbents (with glucose as the carbon source) using a hydrothermal carbonization and chemical activation treatment with ZnCl_2_. The prepared MMT–PCN adsorbent exhibited an excellent performance of 686.94 mg·g^−1^ for the removal of methylene blue (MB). Zhou et al. [26] also reported other novel MMT-supported PCN nanocomposites (derived from cellulose) with an adsorption capacity for the removal of MB of 138.1 mg·g^−1^. Besides the often-used carbon source of glucose and cellulose with totally carbon chains, other N-containing natural polymers such as chitosan (CS) have received increasing attention [27,28,29]. CS is a polycation under acidic conditions, and it can easily intercalate into the MMT layers through the ion exchange process. Its macromolecular backbone contains plenty of amino groups, and the derived PCN definitely contains a high content of the N heteroatom. The introduction of the N heteroatom into carbon lattices effectively modifies its surface physicochemical properties and brings new opportunities to improve their catalytic properties. MMT-supported CS-derived N-doped PCN has been demonstrated as a good Appendix A for Pd metals [30].

However, natural MMT has limited porous structure, layer spacing, and cation exchange capacity. Pillaring modification of MMT can effectively improve these properties [31,32,33,34]. The polycationic precursors for metal oxides, such as Al_2_O_3_, Fe_2_O_3_, TiO_2_, ZrO_2_, Cr_2_O_3_, etc., that can form multi nucleus upon the hydrolysis process, can often be used as pillaring reagents to keep the MMT layers apart and prevent collapse. Among the different kinds of pillared MMT, TiO_2_-pillared MMT [35,36,37,38,39,40] has been proven as one of the most suitable supports for active species as applied in especially gas–solid catalytic reactions, based on its excellent characteristics: expanded interlayer spaces, large surface area, high porous structure, and pore size tunable from the micro- to meso-pore range. Recently, mesoporous TiO_2_-modified active carbon has been used as a novel support for Pd species. Xiao et al. [41] used TiO_2_-supported CS-derived carbon as a support to prepare novel Pd@TiO_2_/N-doped C catalysts, which exhibited high catalytic efficiency and stability for the hydrogenation of vanillin to 2-methoxy-4-methylphenol. It was also found that the carbonization of the CS component within the Al-pillared MMT/CS@Pd catalyst was in favor of further improvement of the comprehensive catalytic performance [42,43]. Based on these studies, it is expected that a combination of porous carbon, TiO_2_, and TiO_2_-pillaring modified MMT should be another novel promising support for Pd species. However, to the best of our knowledge, the preparation of porous carbon modified with both pillared MMT and TiO_2_-supported stabilized Pd (TiO_2_-MMT/PCN@Pd) catalysts applied in liquid–solid heterogeneous coupling reactions has been reported in few studies. 

In this study, a series of TiO_2_-modified MMT-supported PCN-stabilized Pd nanocomposites were synthesized (using CS as a carbon source), followed with fine microstructure characterizations and catalytic performance tests for the Sonogashira coupling reactions between aryl halides and terminal phenyl acetylene. The aim was to evaluate the synergism effects of the combination of PCN, MMT, and TiO_2_ as a composite support for the stabilization of Pd catalysts as applied in liquid–solid organic reactions. The correlations between the microstructure and catalytic performance of the prepared novel catalytic nanocomposites were discussed. In addition, comparisons were made of the prepared novel catalysts with other recent solid-supported palladium catalysts in the Sonogashira reaction [44].

## 2. Results and Discussion

### 2.1. Microstructure of the Catalytic Nanocomposites

The XRD patterns of the starting Na-MMT, TiO_2_-MMT, and TiO_2_-MMT/PCN@Pd nanocomposites are shown in Figure 1a,b. For Na-MMT, the diffraction (001) peak that is attributed to the ordering of the MMT layers locates at the 2*θ* around 7.35°, related to the basal space of 1.20 nm. Considering that the thickness of the MMT layer itself is about 0.96 nm [45,46], the interlayer spacing distance of the starting Na-MMT is about 0.24 nm. The pillaring process involves the cations exchange of the polyhydroxy-Ti^4+^ species exchanged with Na^+^, which props open the silicate layers. Upon high temperature treatment, the intercalated polyhydroxy-Ti^4+^ species are transformed into TiO_2_ nanoparticles, linking permanently with the silicate layers. As a result, for the TiO_2_-MMT and the TiO_2_-MMT/PCN@Pd nanocomposites, the characteristic (001) diffraction peaks become extremely broader and weaker, indicating almost disorderly alignment of the MMT silicate layers after the TiO_2_ modification. Nevertheless, the basal space of 2.05 nm and interlayer spacing distance of 1.09 nm can be derived from the weak peak at 2*θ* of around 4.30°. According to the Lagaly’s method [47,48,49], the bilayer arrangement of the CS chains (PCN precursor) and its derived PCN species occur in the pillared silicate interlayers.

As shown in Figure 1b, for Na-MMT, the diffraction peak at 2*θ* of 19.7° is related to the two-dimensional *hk* indices of (02) and (11), the diffraction peak at 2*θ* of 34.8° is related to the two-dimensional *hk* indices of (13) and (20), the diffraction peak at 26.6° is attributed to quartz, and the diffraction peak at 28.7° is attributed to the silica impurity [50]. For TiO_2_-MMT and all of the TiO_2_-MMT/PCN@Pd nanocomposites, the *hk* reflection peaks at 2*θ* of about 19.7° are related to the two-dimensional MMT layers still present [51]. However, the relative intensity of the diffraction peak obviously decreases. This can be due to the formation of the crystalline TiO_2_ nanoparticles. As shown in Figure 1b, the new diffraction peaks at 2*θ* of about 25.4°, 27.5°, 36.1°, 37.9°, 41.2°, 48.1°, and 54.2° are assigned to the TiO_2_ nanoparticles [52], indicating the presence of mainly anatase TiO_2_ together with some rutile TiO_2_ on the surface of the MMT layers after the calcination process. This phenomenon is quite different from that of the Al-pillared MMT [23,32,33,42], in which most of the intercalated polyhydroxy-Al cations are converted into stable Al_2_O_3_ pillars instead of Al_2_O_3_ nanoparticles dispersed on the surface of the MMT layer.

Further evidence of the successful TiO_2_-pillaring modification can be found in the changes of the FTIR spectra of the starting MMT and the nanocomposites. As shown in Figure 2, after the TiO_2_ pillaring, carbonization and Pd-loading steps, the preservation of characteristic FTIR bands in the region of 400–700 cm^−1^ (465 cm^−1^ assigned to Si-O-Si bending, 527 cm^−1^ assigned to Al-O-Si bending) [53,54,55,56] is observed, indicating that the small building units of the MMT layer are still present. For pure MMT, the peak at 915 cm^−1^ is assigned as the Si-OH vibration. After dehydroxylation in the high-temperature pillaring or carbonization process, the Si-OH vibration peak almost disappears and new peaks are found around 930 and 945 cm^−1^, which can be assigned to the Si-O-Ti vibration, confirming the molecular combination of the TiO_2_ species with the MMT frame.

On the one hand, the pillars successfully derived in the MMT interlayer space should originate from the molecular level of the TiO_2_ species, which are combined on the MMT layer at the molecular level by the Si-O-Ti bonds after the pillaring process (including the polycation precursor intercalation, hydrolysis, and calcination steps). On the other hand, as TiO_2_ also prefers to form nanoparticles with fine crystal structure after calcination at 500 °C, some of the derived TiO_2_ nanoparticles with a larger size than the basal spacing of the pillared MMT might be sandwiched in the multilayer space of the pillared MMT. This results in some irregular stacking of the MMT layers, while the other derived TiO_2_ nanoparticles should be directly dispersed on the external surface of the MMT layers.

Due to the low loading content of Pd and its fine dispersion, the characteristic Pd^0^ crystal diffraction peaks do not appear distinctly. Nevertheless, the Pd content within the catalyst can be supported using the ICP-AES and XPS determination. The percentage of Pd content within the TiO_2_-MMT/PCN@Pd^0^ nanocomposite is determined as about 2% using ICP-AES. Meanwhile, the binding energies of Pd_3d_ are observed as 335.6 eV (assigned to the Pd^0^ species) and 337.2 eV (assigned to the Pd^2+^ species), confirming the presence of the Pd active species [57]. In addition to the Pd_3d_, the XPS spectra of C_1s_, O_1s_, and N_1s_ are shown in Figure 3. For the XPS spectrum of C_1s_ in PCN, it can be deconvoluted into three peaks at 284.4 eV (C atoms on C=C), 285.9 eV (C atoms on C-N and/or C-O), and 288.4 eV (C atoms on C=O), respectively. For the XPS spectrum of N_1s_ in PCN, it can be deconvoluted into four peaks at 399.2 eV (N atoms on pyridinic-N), 400.3 eV (N atoms on pyrrolic-N), 401.6 eV (N atoms on graphitic-N), and 403.0 eV (N atoms on N-O bonds), respectively. For the XPS spectrum of O_1s_, it can be deconvoluted into two peaks at 530.5 eV (O atoms on C-O) and 532.1 eV (O atoms on C=O), respectively. Clearly, the derived N-containing PCN from CS successfully supported on the TiO_2_-MMT is powerfully supported by the XPS results. The Raman shift of the prepared TiO_2_-MMT_60_/PCN_40_@Pd^0^ is illustrated in Figure 4. The peaks at about 1360 cm^−1^ and 1590 cm^−1^ are attributed to disordered and ordered graphite carbon species (the so-called D band and G band), respectively. The derived PCN is mainly composed of disordered carbon species as the *I*_D_/*I*_G_ is found as 4.1.

The N_2_ adsorption–desorption isotherms and corresponding pore size distribution of the starting Na-MMT, TiO_2_-MMT, and TiO_2_-MMT/PCN@Pd nanocomposites are shown in Figure 5. Based on the isotherms, the extracted BET surface area (*S*_BET_), micropore area (*A*_mic_), and total pore volume (*V*_tot_) are listed in Table 1. All of the samples show distinct hysteresis loops at a higher P/P_0_ of 0.4 and considerable adsorption amounts at a low relative pressure, suggesting type IV isotherms with fairly good adsorption capacity. For Na-MMT, the hysteresis loops are assigned to the typical H_3_ type, implying poor mesopores. For the TiO_2_-MMT and TiO_2_-MMT/PCN@Pd nanocomposites, all of the hysteresis loops are assigned to the typical H_4_ type, implying rich, narrow, and slit-like mesopores. The pore size distribution peaks of TiO_2_-MMT are located at about 4 nm (minor peak) and 10 nm (major peak). This suggests that TiO_2_ modification leads to the effective formation of numerous mesoporous layered structures. After hybridizing with PCN and the Pd^2+^ species, the pore size distribution peaks of TiO_2_-MMT/PCN show little shift, indicating that the introduction of these species has limited effects on the mesopore size of the TiO_2_-MMT matrix. However, both in the case of TiO_2_-MMT_80_/PCN_20_@Pd^0^ and TiO_2_-MMT_60_/PCN_40_@Pd^0^, the pore distribution peaks become obviously broader in the range of 10–16 nm. The increase in the larger-sized mesopores is ascribed to the generation of numerous disordered mesopores due to the reduction in the Pd^2+^ species to Pd^0^ nanoparticles. The *S*_BET_ of TiO_2_-MMT (161.1 m^2^/g) is about 13.8 times that of the starting Na-MMT (11.6 m^2^/g). The *V*_tot_ of TiO_2_-MMT (0.37 cm^3^/g) is about 9.2 times that of the starting Na-MMT (0.04 cm^3^/g). This is closely related to the induction of a large number of new stable porous structures after TiO_2_ pillaring, which is in good agreement with the XRD results. However, after the loading of the PCN and the Pd species, it is observed that the *S*_BET_ and *V*_tot_ show a reasonably decrease. For the TiO_2_-MMT_80_/PCN_20_@Pd^0^ series, *S*_BET_ decreases to 137.3 m^2^·g^−1^ (TiO_2_-MMT_80_/PCN_20_), 114.9 m^2^·g^−1^ (TiO_2_-MMT_80_/PCN_20_@Pd^2+^), and 108.9 m^2^·g^−1^ (TiO_2_-MMT_80_/PCN_20_@Pd^0^), respectively; *V*_tot_ shows a little decrease to 0.37 cm^3^/g (TiO_2_-MMT_80_/PCN_20_), 0.36 cm^3^/g (TiO_2_-MMT_80_/PCN_20_@Pd^2+^), and 0.27 cm^3^/g (TiO_2_-MMT_80_/PCN_20_@Pd^0^), respectively. For the TiO_2_-MMT_60_/PCN_40_@Pd^0^ series, *S*_BET_ decreases to 124.2 m^2^·g^−1^ (TiO_2_-MMT_60_/PCN_40_), 121.1 m^2^·g^−1^ (TiO_2_-MMT_60_/PCN_40_@Pd^2+^), and 89.6 m^2^·g^−1^ (TiO_2_-MMT_60_/PCN_40_@Pd^0^), respectively. *V*_tot_ shows a little decrease to 0.33 cm^3^/g (TiO_2_-MMT_60_/PCN_40_), 0.30 cm^3^/g (TiO_2_-MMT_60_/PCN_40_@Pd^2+^), and 0.27 cm^3^/g (TiO_2_-MMT_60_/PCN_40_@Pd^0^), respectively. Nevertheless, the adsorption capacity and porosity of such novel TiO_2_-MMT/PCN@Pd^0^ is superior to the recently prepared adsorbents or heterogeneous catalysts using TiO_2_-pillared MMT or MMT/PCN composites as a matrix [35,36,37,38,39,40]. As confirmed in the adsorption tests for the rhodamine B dye at room temperature (as shown Figure 6), both TiO_2_-MMT_80_/PCN_20_@Pd^0^ and TiO_2_-MMT_60_/PCN_40_-Pd^0^ show fairly good adsorption removal efficiency for the dye molecules (only 20 min reaching the equilibrium adsorption removal). Obviously, the former (92.1%) is better than the latter (79.6%). This might be due to the higher surface area of TiO_2_-MMT_80_/PCN_20_@Pd^0^ than TiO_2_-MMT_60_/PCN_40_-Pd^0^. The highly porous structure and high surface area performance should be in favor of excellent catalytic performance of the resultant nanocomposites. 

The HRTEM-EDX images of the starting Na-MMT, TiO_2_-MMT/PCN support and TiO_2_-MMT/PCN@Pd nanocomposites are shown in Figure 7. The starting MMT exhibits a layered structure with a regular stacking and closing interlayer distance (Figure 7A). For TiO_2_-MMT, the loaded TiO_2_ exist in three forms. Firstly, increasing of the interlayer distance and contrast is observed, indicating successful pillaring of the TiO_2_ species. However, the TiO_2_ pillars seem difficult to identify using HRTEM, which can be due to the molecular level combing of the TiO_2_ species with the MMT framework of the Si-O-Ti bonds. Similar phenomenon are also reported in other studies on pillared MMT characterized with TEM [58,59,60]. Secondly, it is observed (as marked with a blue rectangle) that some TiO_2_ nanoparticles sized about 5 nm are clipped in multilayer spaces of the MMT, causing an obvious disordered stacking of the MMT layers. Thirdly, many of the other visible TiO_2_ nanoparticles disperse directly on the external surface and/or edge of the MMT layers. After immobilization of the Pd species, it is observed that some additional nanoparticles (with sizes below 2 nm, as marked with red circles) are clipped in the space of the adjacent MMT layers, which can be attributed to the Pd^0^ species. Unfortunately, for the low contrast, the successful loading of the PCN species is difficult to identify with the HRTEM images. Nevertheless, with the element mapping and weight ratio results from the HRTEM-EDX, the successful loading of the PCN and Pd species is further confirmed. Based on the HRTEM-EDX results, the actual chemical composition of the TiO_2_-MMT_60_/PCN_40_@Pd^0^ may be estimated as 47% of MMT, 41% of TiO_2_, 11% of PCN, and 1% of Pd, respectively. As shown in Appendix A, the success incorporation of the PCN and Pd species can also be powerfully supported by the element’s composition of the TiO_2_-MMT_60/_PCN_40_@Pd^0^ using SEM-EDX. Clearly, the HRTEM-EDX results are consistent with the XRD, FTIR, and N_2_ adsorption–desorption results.

### 2.2. Positron Annihilation Characteristics of the Catalytic Nanocomposites

According to the microstructure characterization results above, the textile structure of the nanocomposite can be illustrated in Figure 1. The mesoporous structure (using the N_2_ adsorption–desorption isotherms and adsorption-removal test), interlayer spacing (using XRD), morphologies (using HRTEM), and compositions (using HRTEM-EDX, Raman spectroscopy, FTIR) are successfully revealed. However, the sub-nano level microstructure, such as the molecular packing information of the derived TiO_2_ pillars, PCN, and active Pd species inside the TiO_2_-modified MMT layer space still lacks essential evidence. Positron annihilation lifetime spectroscopy (PALS) has recently been proven as one of the most highly sensitive methods to detect microstructure information at the sub-nano level [61,62,63,64,65]. In the interlayer space with low electron density, in addition to free positron annihilation, some of the thermalized positrons can be trapped with electrons to form a bound state called positronium (Ps), and then annihilate. Ps has two states, *o*-Ps (spin parallel) and *p*-Ps (spin antiparallel), depending on the spin orientations of the positrons and bound electrons. In a vacuum, the intrinsic lifetimes of *p*-Ps and *o*-Ps are 0.125 ns and 142 ns, respectively. In molecular solids, *o*-Ps will form interactions with the electrons in the surrounding medium and undergo pick-up annihilation, resulting in a reduction in the lifetime of 1 to several ns. Using a suitable quantum mechanical model [66,67], the *o*-Ps lifetime is correlated to the microdefect size. It can then be used as a sensitive probe to detect the local microdefect structure. 

In such TiO_2_-MMT/PCN@Pd nanocomposite systems, as shown in Figure 1, *o*-Ps annihilation should mainly occur in the molecule-stacking gaps in all of the involved molecular substrates in the interlayer space of the MMT with low electron density, such as the PCN molecules, Pd species, TiO_2_ pillars, and MMT layers. Using the quantum mechanical model in Equation (1) [68,69], the width of the cuboidal defects size can be estimated, where *τ*_3_ refers to the longest lifetime, l refers to the width of the cuboidal defects size, and Δ*l* (=0.17 nm) refers to the thickness of the fitted empirical electron layer. Therefore, *o*-Ps can be used as a highly sensitive probe to detect the molecule packing and interfacial interactions in the interlayer space of the MMT layers. The microdefects fraction, *f*, i.e., a combination of o-Ps intensity *I*_3_ with microdefects volume (*V* = *l*_3_), can be calculated according to Equation (2), where C is a constant. To simplify, the apparent microdefects fraction (*f*_app_) is often used to determine the variation trends.
(1)$$\tau_{3}=0.5{\left[1-\left(\frac{l}{l+2\Delta l}+\frac{1}{\pi }\sin\frac{\pi l}{l+2\Delta l}\right)\right]}^{-1}$$
*f* = C*Vl*_3_ or *f*_app_ = *Vl*_3_
(2)

As shown in Table 2, the positron annihilation spectra of the nanocomposites are fitted well in the three-lifetime fitting with the LT-9 program. For all of the samples, the first lifetime component of ***τ*_1_** and its intensity of *I*_1_ can be attributed to a combination of *p*-Ps annihilation and free positron annihilation. The second lifetime component of ***τ*_2_** and its intensity of *I*_2_ can be attributed to a combination of free positron annihilation and some trapped positron annihilation in the microdefects of the crystalline MMT layers. The third long lifetime component of ***τ*_3_** and its intensity of *I*_3_ can be attributed to the *o*-Ps pick-off annihilation in the interlayer spaces of the TiO_2_-MMT_60_/PCN_40_@Pd^0^ nanocomposites as illustrated in Figure 1. In the confined nanospace between the neighboring layers, the *o*-Ps will be mainly trapped in the molecular packing gaps of the involved substrates, such as the MMT layer, PCN species, Pd species, and TiO_2_ species. Using the cuboidal microdefects model in Equation (1), the size of the microdefects inside the interlayer space can be calculated. The microdefects size (*l*) inside the interlayer space of pure MMT can be calculated as 0.290 nm. After TiO_2_ pillaring, modification, and further PCN derivation, the microdefects size of *l* increases to 0.307 nm (TiO_2_-MMT) and 0.311 nm (TiO_2_-MMT_60_/PCN_40_). This increase in size can be related to the increase in interlayer space caused by the effective high-temperature process during the TiO_2_ pillaring and carbonization steps. After Pd^2+^ immobilization and further reduction in the nano Pd^0^ species, the microdefect size of *l* decreases to 0.308 nm (TiO_2_-MMT_60_/PCN_40_@Pd^2+^) and 0.299 nm (TiO_2_-MMT_60_/PCN_40_@Pd^0^), which confirms that the interlayer space becomes more crowded after the Pd species are successfully incorporated into the interlayer space of the modified MMT. Moreover, it is observed that the both *I*_3_ and *f*_app_ of the TiO_2_-MMT_60_/PCN_40_@Pd nanocomposites are higher than the starting MMT, TiO_2_-MMT, and TiO_2_-MMT_60_/PCN_40_ supports. This means that the nanocomposite still has high microdefect features though the interlayer spaces is more crowded. As microdefects can be used as active sites for catalytic reactions, a high comprehensive catalytic performance of the TiO_2_-MMT_60_/PCN_40_@Pd nanocomposites is expected.

### 2.3. Performances of the Catalytic Nanocomposites Applied in Sonogashira Reactions

The Sonogashira reaction, usually referring to the cross-coupling reaction of terminal alkyne with aryl halides, is an extremely valuable type of reaction to form C-C (sp-sp_2_) bonds [70,71,72]. It has been broadly used in the field of synthesis of functional molecules of natural products, biological active features, pharmaceuticals, heterocycles, conducting polymers, and liquid polymer substrates, etc. Firstly, the catalytic performances of the TiO_2_-MMT/PCN@Pd nanocomposites are evaluated with a typical model Sonogashira reaction of iodo benzene and phenyl acetylene. The Pd^0^ species supported on TiO_2_, PCN, TiO_2_/PCN, MMT/PCN, and TiO_2_-MMT are prepared and their N_2_ adsorption performances are shown in Appendix A. As shown in Appendix A, the model Sonogashira reaction is found to be difficult to perform without any Pd catalyst. However, with the presence of different Pd catalysts, such as TiO_2_@Pd^0^, PCN@Pd^0^, TiO_2_/PCN@Pd^0^, MMT/PCN@Pd^0^, and TiO_2_-MMT@Pd^0^, the reaction can be fairly well catalyzed with good yields (Appendix A). As shown in Figure 8A, for each reaction time interval, it is found that the reaction yields catalyzed by TiO_2_-MMT_80_/PCN_20_@Pd^0^ are higher than that catalyzed by TiO_2_-MMT_60_/PCN_40_@Pd^0^, which should be mainly attributed to its higher adsorption capability [73,74]. The recyclability of the catalysts is further evaluated as applied in the model reaction. At each reaction time interval, hardly any improvements of the yields are observed after a hot filtration out of both catalysts, confirming the high heterogeneity. As shown in Figure 8B, for maintaining the yield higher than 70%, TiO_2_-MMT_80_/PCN_20_@Pd^0^ and TiO_2_-MMT_60_/PCN_40_@Pd^0^ can recycle for 16 runs and 19 runs, respectively. Obviously, as compared with the recyclable runs of TiO_2_-Pd^0^ (six runs), PCN-Pd^0^ (ten runs), TiO_2_/PCN-Pd^0^ (eleven runs), MMT/PCN-Pd^0^ (eight runs), and TiO_2_-MMT-Pd^0^ (six runs) (as shown in Appendix A), a combination of the PCN, MMT, and TiO_2_ into a composite support Pd metal catalyst shows effective synergism improvements in the recyclability of the resultant TiO_2_-MMT/PCN@Pd^0^ catalytic nanocomposites. Moreover, as shown in Figure 8C and Appendix A, TiO_2_-MMT/PCN@Pd shows higher stability than most of the recently developed Pd heterogeneous catalysts for Sonogashira reactions with similar reaction conditions, including Pd loaded on modified TiO_2_ (h-Fe_3_O_4_@TiO_2_-NH_2_/Pd or (TiO_2_@Pd (II)[PATA-NH2]) [75,76], MMT (K10@Pd(II)APTES or MMT/CS@Pd, Cu) [77,78], or carbon-based supports (CHT@Pd or Hal-pDA-NPC@Pd) [79,80]. The excellent chelation, stability, and adsorption of TiO_2_, MMT, and PCN have been well combined into a composite material, resulting in an effectively synergetic performance improvement in the case of the TiO_2_-MMT/PCN@Pd catalyst. The higher stability (three more recyclable runs) of TiO_2_-MMT_60_/PCN_40_@Pd^0^ than TiO_2_-MMT_80_/PCN_20_@Pd^0^ may be attributed to its higher content of the PCN, which has reasonably stronger chelation with the Pd species as compared with inorganic TiO_2_ or MMT. Hence, further evaluation of the catalyst performance was mainly focused on the case of TiO_2_-MMT_60_/PCN_40_@Pd^0^ with a higher content of PCN. As tracked using the ICP-AES assay, about 93% and 81% of the Pd species retained in the recovered TiO_2_-MMT_60_/PCN_40_@Pd^0^ for five runs and ten runs, respectively. To clarify the reason for the Pd leaching, the recovered catalyst was further characterized with PALS.

As shown in Figure 8D, distinct differences are observed for the TiO_2_-MMT_60_/PCN_40_@Pd^0^ after recycled for different times. As compared with the fresh TiO_2_-MMT_60_/PCN_40_@Pd^0^ nanocomposite, many more counts of o-Ps annihilations with a longer lifetime (>2.5 ns) are observed for the PALS spectra of the recycled nanocomposites. This indicates that the recycled nanocomposites may contain more than one long-lifetime component as the usual molecular solids. Therefore, the positron annihilation spectra of the recycled nanocomposites have been refitted in four-lifetime rather than the usual three-lifetime fitting. The PALS spectrum of fresh TiO_2_-MMT_60_/PCN_40_@Pd^0^ catalyst has been also fitted in the four-lifetime component for comparison. As shown in Table 3, for the spectrum of fresh TiO_2_-MMT_60_/PCN_40_@Pd^0^ nanocomposite, we obtained *τ*_1_ = 0.249 ns, *τ*_2_ = 0.372 ns, *τ*_3_ = 0.472 ns, and *τ*_4_ = 2.11 ns with the relative intensities *I*_1_ = 71.3%, *I*_2_ = 8.8%, *I*_3_ = 18.0%, and *I*_4_ = 1.9%, exhibiting three short-lifetime components (<0.5 ns) and one long-lifetime component (>1 ns). The first two short-lifetime components of *τ*_1_ and *τ*_2_ can be attributed to the *p*-Ps decay and free positron annihilations. The third short-lifetime component of *τ*_3_ can be attributed to free positron annihilations and some trapped positron annihilations in the microdefects of the MMT crystalline layer and the TiO_2_ nanoparticle crystals. The long-lifetime component of *τ*_4_ can be attributed to *o*-Ps pick-off annihilations in the molecules stacking gaps of all of the involved molecular substrates, such as the PCN molecules, Pd species, TiO_2_ nanoparticles, and MMT layers. The mean microdefects size *l* (from *τ*_4_) of the fresh TiO_2_-MMT_60_/PCN_40_@Pd^0^ is estimated as 0.277 nm. Clearly, this size value is even smaller than the microdefects size l of 0.299 nm that is obtained from the three-lifetime fitting (as shown in Table 2). This indicates that the PCN and Pd species are tightly encaged in the nanospace of the TiO_2_-pillared MMT layers and three-lifetime fitting is more reasonable than four-lifetime fitting for fresh TiO_2_-MMT_60_/PCN_40_@Pd^0^ nanocomposites.

Unlike the fresh catalyst, after recycled for five runs or ten runs, two long-lifetime components (>1 ns) of *τ*_3_ and *τ*_4_ are observed. For the recycled TiO_2_-MMT_60_/PCN_40_@Pd^0^ catalytic nanocomposite recycled for five runs, besides the first long-lifetime component of *τ*_3_ = 2.13 ns (close to the long-lifetime of τ4 of the fresh catalytic nanocomposite), the second long-lifetime of *τ*_4_ is observed as 7.2 ns (with the relative intensities of *I*_4_ = 0.26%). This indicates that *o*-Ps pick-off annihilations occur in other much larger-sized microdefects. The microdefect sizes of the five-runs recycled TiO_2_-MMT_60_/PCN_40_@Pd^0^ catalyst are estimated as 0.279 nm and 0.616 nm. Similarly, for the recycled TiO_2_-MMT_60_/PCN_40_@Pd^0^ catalyst recycled for ten runs, the microdefects sizes are estimated as 0.274 nm and 0.681 nm. This suggests that PALS is highly sensitive to the development of microdefects and provides direct evidence for the induction of some larger-sized microdefects during sequential recycling. Meanwhile, the relative intensities of *o*-Ps annihilation (*I*_3_ + *I*_4_) of the five-runs recycled catalyst (*I*_3_ + *I*_4_ = 3.3%) and ten-runs recycled catalyst (*I*_3_ + *I*_4_ = 4.2%) are obviously higher than that of fresh catalyst (*I*_4_ = 1.9%), indicating that more microdefects form after the continuous recycling process. Although larger-sized microdefects have low intensities, some of them may provide leaching channels for load molecules, including active Pd species.

Moreover, the novel TiO_2_-MMT_60_/PCN_40_@Pd^0^ nanocomposite can be successfully extended to a range of substituted aryl halides coupling with phenyl acetylenes. As shown in Table 4, the reactions of the aryl iodides substituted with electron-donating groups of -CH_3_ at ortho (entry 2), meta (entry 3), or para (entry 4) positions with phenylacetylene achieved excellent yields of 92–99%. Furthermore, the reactions of the aryl iodides substituted with electron-withdrawing groups of -F, -Br, and -Cl at ortho (entry 5), meta (entry 6), or para (entry 7) positions with phenylacetylene achieved excellent yields of 93–99%. For the entries 8–12, the reactions of the substituted aryl bromides with phenyl acetylene can be still effectively catalyzed with the novel TiO_2_-MMT_60_/PCN_40_-Pd^0^ nanocomposite with desirable yields of 59–72%. In addition, the novel catalysts can efficiently catalyze the large-sized reactants for the coupling of iodo naphthalene and iodo fluorene with phenyl acetylene (entry 13, 14), indicating the high feasibility of molecular size. Conclusively, the TiO_2_-MMT_60_/PCN_40_@Pd^0^ nanocomposite developed in this study shows similarly high catalytic efficiency as compared with recently reported heterogeneous Pd catalysts for Sonogashira reactions with similar reaction conditions [23,75,76,77,78,79,80]. The good dispersion and tight loading of the Pd species in the highly porous, stable, and amphiphilic TiO_2_-pillared MMT-supported PCN matrix is likely the main reason for this high catalytic efficiency.

## 3. Materials and Methods

**Materials:** G-105 type Na+-MMT clay from Nanocor Co., USA, with cationic exchange capacity of 145 meq/100 g, was used as the starting material. Chitosan (CS) from Zhejiang Aoxing Biotechnology Co., Ltd., viscosity molecular weight of 1.2 × 10^5^ and deacetylated degree of 95%, was used as the precursor of porous carbon (PCN). All of the chemical reagents and reactant molecules involved in Sonogashira reactions used in this study were of analytical grade without further purification.

**Preparations:** The TiO_2_-modified MMT was prepared using similar processes as in recent studies [39]. TiCl_4_ was added dropwise into 2 mol/L HCl solution under mechanical stirring and ice–water (0 °C) bath. The mixture was then diluted to reach the concentration of 0.6 mol/L of H^+^ and 0.82 mol/L of Ti^4+^, respectively. The pillaring solution was aged for 12 h at 25 °C prior to its use. The pillaring solution was added dropwise into 10 wt% Na^+^-MMT clay aqueous suspension, to reach the Ti^4+^/Na^+^-MMT clay ratio of 20 mmol/1 g. The mixed suspension was kept mechanically stirring for 6 h at 60 °C, and then it was centrifuged and washed with deionized water to neutral. After naturally drying, the mixture was calcined in a tubular muffle furnace (BTF-1600C, Anhui BEQ Equipment Technology Co., Ltd., Hefei, China) for 3 h at a temperature of 500 °C in N2 atmosphere. A total of 0.5 and/or 1.33 g of CS was dissolved in 200 mL of 2 wt% CH_3_COOH solution. An amount of 5 mL of Na_2_PdCl_4_ solution (containing 0.09 mmol of Pd) was added dropwise into the CS solution. A total of 2 g of the resultant TiO_2_-modified MMT was added into 200 mL of the above CS or CS-Pd^2+^ complex solution, and continuously stirred at 60 °C (water-bath heating) for 10 h. The TiO_2_-MMT/CS or TiO_2_-MMT/CS@Pd^2+^ products were separated from the suspension by centrifugation. After washing with deionized water to neutral and naturally drying, the TiO_2_-MMT/CS or TiO_2_-MMT/CS@Pd^2+^ was carbonized into TiO_2_-MMT/PCN or TiO_2_-MMT/PCN@Pd^2+^ at 800 °C for 4 h under N_2_ atmosphere in the tubular muffle furnace. Before catalytic use, the resultant TiO_2_-MMT/PCN@Pd was further reduced with ethylene glycol at 80 °C for about 1 h to obtain TiO_2_-MMT/PCN@Pd^0^. The nanocomposites were labelled according to the mass ratio of the TiO_2_-MMT to CS (i.e., 80/20, and 60/40), such as TiO_2_-MMT_80_/PCN_20_, MMT_60_/PCN_40_, TiO_2_-MMT_80_/PCN_20_@Pd^2+^, TiO_2_-MMT_80_/PCN_20_@Pd^0^, TiO_2_-MMT_60_/PCN_40_@Pd^2+^, andTiO_2_-MMT_60_/PCN_40_@Pd^0^, respectively. For comparing the catalytic performances, Pd^0^ immobilized on TiO_2_, PCN, TiO_2_/PCN, MMT/PCN, TiO_2_-MMT were also prepared, and the preparing process is shown in the Appendix A.

**Characterizations:** X-ray diffraction (XRD) patterns of the samples with 2*θ* from 2–10° (small angle) and 10–60° (wide angle) were recorded using a Brucker D8 Advance (Bruker Corporation, USA) with a scanning rate of 1°/min, employing Cu Kα radiation (*λ* = 0.154 nm). N_2_ adsorption–desorption isotherms of the samples were recorded using a Micromeritics TriStar II 3020 apparatus (Micromeritics Company, USA) at the liquid N_2_ temperature of 77 K. Before the N_2_ adsorption–desorption isotherms test, the samples were degassed at 200 °C for 6 h. The BET measurements were repeated 3 times. Morphology of the samples was observed using a JEM-2100 F high-resolution transmission electron microscope HRTEM (JEOL Ltd., Tokyo, Japan) equipped with an energy-dispersive X-ray-spectroscope (Oxford EDX system). The HRTEM samples were firstly dispersed in ethanol in an ultrasonic bath for 5–10 min, and then deposited on a Cu grid and dried at room temperature. The samples were also observed with a scanning electron microscope SEM (JEM-6360, JEOL Ltd. Japan) equipped with an energy-dispersive X-ray spectroscope (Oxford EDX system). Raman shifts of the samples were recorded using a DXR Raman Imaging Microscope (Thermo Scientific, USA) with excitation laser wavelength of 532 nm. Fourier transform infrared spectroscopy (FTIR) analysis of the samples was performed with a Thermo-Nicolet 470 FT-IR (USA) spectrometer (scans: 75 times/second; resolution: better than 0.09 cm^−1^) under the wave number ranging from 4000 to 400 cm^−1^, using KBr pressed-disk method. The mass ratio of the sample/KBr is about 1/100. Binding energies of the elements were determined using a Thermo Scientific ESCALAB 250Xi X-ray photoelectron spectrometer (USA). Pd content of the catalyst was determined with a Leemann ICP-AES Prodigy XP inductively coupled plasma-atomic emission spectrometer (USA). Samples for ICP-AES experiments were pre-treated with a mixed solution of concentrated HCl/fuming HNO_3_ (3/1) and then diluted. Positron annihilation lifetime spectroscopy analysis of the samples was performed with an EG&G ORTEC fast–slow system (US). ^22^Na (16 μCi) was used as positron source and the time resolution of the measurements was 190–220 ps. The total counts of lifetime spectrum of each sample were at least 2 × 10^6^. Lifetime spectrum of each sample was resolved with the Lifetime 9.0 program.

**Catalysis applications:** The activity of the catalysts for the Sonogashira reactions was tested. A total of 1 mmol of aryl halide, 1.2 mmol of alkyne, TiO_2_-MMT/PCN@Pd^0^ nanocomposites (containing about 2 μmol of Pd^0^), 3 mmol of CH_3_COOK base, 0.2 mL of ethylene glycol, and 5 mL of dimethyl sulfoxide (DMSO) solvent was mixed in a 50 mL reaction tube with magnetic stirring for 3 h at 110 °C (oil-bath heating). The 1HNMR spectra of each reaction product were determined with a Brucker 400-Hz NMR to verify the chemical structures, which were consistent with our previous studies [23,38,42]. Reaction yields of the reactions were based on the GC/MS measurements. The recyclability of the catalyst was evaluated with a model reaction of iodo benzene with phenyl acetylene. After each run, catalysts were separated and rinsed with ethanol. Afterwards, the recycled catalysts were reused in the next model reaction run.

**Adsorption tests:** Prior to the test, a calibration curve was obtained using the standard rhodamine B solution with known concentrations of 1, 2, 4, 6, 8, 10, 12, 14, 16 mg/L determined with a UV-vis spectrophotometer (UV-754, Shanghai) at an absorbance wavelength of 554 nm. At room temperature, 0.05 g TiO_2_-MMT_80_/PCN_20_@Pd^0^ or TiO_2_-MMT_60_/PCN_40_@Pd^0^ nanocomposite was added into the 100 mL of 50 mg/L rhodamine B solution under stirring. After specific time intervals of 10, 20, 30, 40, 50, 60 min, the sample solutions were filtered to determine the residual concentrations with UV-vis spectrophotometer using a calibration curve. The dye removal rate at time *t* (%) was calculated as (*C*_0_-*C*_t_)/*C*_0_ × 100%, where *C*_0_ was the initial concentration of the dye solution, and *C*_t_ was the concentration of the dye solution at time *t*.

## 4. Conclusions

In summary, a combination of three catalyst supports of MMT clay, TiO_2_, and PCN into a hybrid system for stabilizing Pd species achieved a synergistic improvement of the comprehensive performance of the catalyst for Sonogashira reactions. The microstructure of the TiO_2_-MMT_60_/PCN_40_@Pd^0^ nanocomposite was carefully characterized using XRD, N_2_-adsorption, and TEM, etc. The TiO_2_-pillaring modification can effectively improve the mesoporous structure of MMT. In situ PCN derived from CS was well encaged in the enlarged layer spaces of the MMT. The PALS analysis sensitively detected the development of sub-nano level microdefects during the preparation and recycling process of the catalytic nanocomposites. The successful incorporation of the TiO_2_, Pd, and PCN species within the interlayer nanospace of MMT was well reflected by the changes in the microdefects’ information from the PALS analysis. In addition, after continuous long-term recovery, many newly developed large-sized microdefects were sensitively detected in the PALS analysis. Correspondingly, the four-lifetime fitting was more appropriate than the usual three-lifetime fitting for the analysis of the recovered catalysts. This study provided direct and instructive evidence for the decrease in the catalytic performance of recycled heterogeneous catalysts after long-term service from the aspect of sub-nano level microdefects.

## Data Availability

Data are available from authors based on reasonable requirement from readers.

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
