# Peer review of "TiO2-Modified Montmorillonite-Supported Porous Carbon-Immobilized Pd Species Nanocomposite as an Efficient Catalyst for Sonogashira Reactions"

_molecules, 2023, doi:10.3390/molecules28052399_

Round 1
Reviewer 1 Report
The paper presents the catalytic efficacy for Sonogashira reactions of TiO2 modified montmorilllonite supported porouss carbon immobilized Pd species nanocomposite. The research presented excellent data to support the conclusions of the study and the discusssion of the results is complete. However, there are sentences that are not easy to comprehend at first read, due to the writing style, hence, an English language revision is necessary to improve the readability of the article.
Author Response
We appreciate the positive comments from this reviewer and we are also very enthusiastic about these encouraging results. We followed the recommendations and revised the manuscript accordingly. Many sentences have been rewritten for better presentation. All the detail revisions have been highlighted in red in the revised manuscript.
Reviewer 2 Report
TiO2 modified montmorillonite supported porous carbon im-mobilized Pd species nanocomposite as an efficient catalyst for Sonogashira reactions from Chen et al. it is well crafted and structured. Even so, the manuscript needs to be improved.
The abstract should be more substantive and should contain the most important essence of the manuscript.
The introduction is well prepared, it contains numerous comparisons with published works. Some may still be considered. https://www.mdpi.com/2073-4344/8/5/202
Line 58-65: This research has dealt with this in detail. https://www.sciencedirect.com/science/article/pii/S0169131719301413
Line 88: Why do the authors think about TiO2?
Line 116: This basal distance is characteristic of sodium montmorillonite, which is usually found at 1.24 nm. What is the interaction between Na+ and TiO2?
Line 116-124: Based on Lagaly's method, you should mention mono, bi and pseudotrilayer arrangement with your samples.
Line 125: Fig.1 a) In the case of significant noise, it looks like an extremely amorphous phase.
Line 154: FTIR spectra should be terminated on both sides. Also, usually FTIR spectra orient from 4000-400 cm-1 from right to left.
Line 234: Measurement deviations should be reported. How many times were the BET measurements repeated?
Line 440: That is an extremely high CEC. What is the chemical composition of this synthetic nano-MMT?
Line 485: What was the sample/KBr ratio? How many scans and what resolution did you use for the FTIR measurements?
Overall, I think the manuscript has a perspective, the novelty that should be declared is most important.
Author Response
1. TiO2 modified montmorillonite supported porous carbon im-mobilized Pd species nanocomposite as an efficient catalyst for Sonogashira reactions from Chen et al. it is well crafted and structured. Even so, the manuscript needs to be improved.
Response:We appreciate the positive comments from this reviewer and we are also very enthusiastic about these encouraging results. We followed the recommendations and revised the manuscript accordingly. Many sentences have been rewritten for better presentation. All the detail revisions have been highlighted in red in the high-lightened version of the revised manuscript.
2. The abstract should be more substantive and should contain the most important essence of the manuscript.
Response:The abstract has been rewritten according to the reviewer’s suggestion. The essential novelty points have been added. The catalytic results of the nanocomposites have been listed.
3. The introduction is well prepared, it contains numerous comparisons with published works. Some may still be considered. https://www.mdpi.com/2073-4344/8/5/202
Response:The paper is cited and discussed as ref. 44 according to the reviewer’s suggestion.
4. Line 58-65: This research has dealt with this in detail. https://www.sciencedirect.com/science/article/pii/S0169131719301413
Response:The paper is cited and discussed as ref. 24 according to the reviewer’s suggestion.
5. Line 88: Why do the authors think about TiO2?
Response:The excellent characteristics of TiO2 pillaring modification have been rediscussed: expanded interlayer spaces, large surface area, high porous structure, and pore size tunable from micro- to meso-pore range.
6. Line 116: This basal distance is characteristic of sodium montmorillonite, which is usually found at 1.24 nm. What is the interaction between Na+ and TiO2?
Response:The basal distance from the XRD measurement in this work is 1.20 nm, which is close to 1.24 nm. The interaction between Na+ and TiO2 is discussed as follows. The pillaring process involves the cations exchange of the polyhydroxy-Ti4+ species exchanged with Na+, which props open the silicate layers. Upon high temperature treatment, the intercalated polyhydroxy-Ti4+ species are transformed into TiO2 nano-particles, linking permanently with the silicate layers.
7. Line 116-124: Based on Lagaly's method, you should mention mono, bi and pseudotrilayer arrangement with your samples.
Response:According to the Lagaly’s method, bilayers arrangement of the CS chains (PCN precursor) and its derived PCN species in the pilled silicate interlayers occurs. New ref. 48, 49 have been cited.
8. Line 125: Fig.1 a) In the case of significant noise, it looks like an extremely amorphous phase.
Response:We agree with the reviewer. New discussion could be found as follows. As a result, for the TiO2-MMT and the TiO2-MMT/PCN@Pd nanocomposites the char-acteristic (001) diffraction peaks become extremely broader and weaker, indicating al-most disorderly alignment of the MMT silicate layers after TiO2 modification. Never-theless,referring to the weak peak at 2θ of around 4.30 ° , the basal space of 2.05 nm and interlayer spacing distance of 1.09 nm can be derived.
9. Line 154: FTIR spectra should be terminated on both sides. Also, usually FTIR spectra orient from 4000-400 cm-1 from right to left.
Response:FTIR spectra have been modified according to the reviewer’s suggestion.
10. Line 234: Measurement deviations should be reported. How many times were the BET measurements repeated?
Response:Measurement deviation have been added. The BET measurements repeat for 3 times.
11. Line 440: That is an extremely high CEC. What is the chemical composition of this synthetic nano-MMT?
Response:CEC of the MMT in this study is 145 meq/100 g. Based on HRTEM-EDX results, the actual chemical composition of the TiO2-MMT60/PCN40@Pd0 might be estimated as 47% of MMT, 41% of TiO2, 11% of PCN, and 1% of Pd, respectively.
12. Line 485: What was the sample/KBr ratio? How many scans and what resolution did you use for the FTIR measurements?
Response:The mass ratio of the sample/KBr is about 1/100. scans: 75 times/second; resolution: better than 0.09 cm-1
13. Overall, I think the manuscript has a perspective, the novelty that should be declared is most important.
Response:Thanks again for the positive comments by the reviewer. We hope that our efforts in improving the paper can meet the requirements of reviewers.
Round 2
Reviewer 2 Report
The authors satisfactorily answered on all comments and also improved manuscript. Manuscript can be accepted.